# Travel Behaviour Insights among Geotourists in Serbia—Case Study of Zaječar District

Miloš Marjanović *, Nemanja Tomić, Aleksandar Antić  and Tijana Tomić

Department of Geography, Tourism and Hotel Management, Faculty of Sciences, University of Novi Sad, Trg Dositeja Obradovića 3, 21000 Novi Sad, Serbia; airtomic@gmail.com (N.T.); a.antic994@gmail.com (A.A.); dordevictijana@gmail.com (T.T.)

* Correspondence: milos.marjanovic@mail.com

**Abstract:** Geotourism is a rapidly growing market for tourism, and has gained huge popularity worldwide. Zaječar district is located in Eastern Serbia, and this area is famous for many attractive geotourism features that seek to be presented to the global tourism market. This article aims to present geotourist typology models based on their motivation and travel behaviour. A questionnaire survey was conducted with 194 respondents who visited the geosites of Zaječar district or have the intention to visit them. The data was processed by an exploratory factor analysis, one-way ANOVA, the *t*-test for independent samples, and multiple regression analyses for in-depth investigations and statistical validation of the findings. The results present three typology models of geotourists based on their motivation to visit geosites (health and relaxation, education and curiosity, socialisation), and three typology models of geotourists based on their travel behaviour (active behaviour, passive behaviour, individual behaviour). The analysis also revealed that motives significantly predict tourist behaviour. Also, this study shows that respondents (tourists) have a positive attitude towards local communities, and emphasise their importance for geotourism development. These findings could be helpful for policy managers and all other interested parties to create strategies and tourism products according to the needs of the potential geotourism market.

**Keywords:** motivation; travel behaviour; local community; geotourists; geotourism; tourist habits



## 1. Introduction

In recent years, a fast-growing number of protected areas has been recorded worldwide, and the tourism development within these areas has become an important activity [1]. Geotourism represents a form of tourism which puts focus on geological and geomorphological peculiarities, both in-situ and ex-situ, providing a possibility for their interpretation and conservation, education of the tourism market [2], and emphasising sustainable tourism development and conservation, which provide benefits for communities and the economy [3]. The geotourism market has been growing rapidly worldwide over the years, and it is estimated that it will continue to grow [4]. A large body of literature deals with the presentation of geoheritage, as well as its assessment for tourism purposes, or sustainable use, however, it is also important to know what motivates tourists to visit a particular destination. Emerging trends in nature-based tourism forced policymakers and other stakeholders to constantly follow the preferences of the target market sector, so they can adjust their tourism products towards new travel trends, and be competitive in a turbulent tourism market. There is various research providing insight into what motivates tourists to travel, as this information is very important to stakeholders for market segmentation. The segmentation of the tourism market is also very useful for the sustainable use of limited resources. On the other hand, there is a limited number of articles related to the motivation of geotourists [5–8], their travel behaviour [9–11], and their attitude towards the local community [12].

Understanding why people travel has been a research topic since the middle of the twentieth century, as it helps to predict travel behaviour [13]. In tourism research, a commonly used theory is Crompton's push-and-pull motivation theory [14]. Later, Crompton's motivation theory, and other articles related to motivation [15,16], inspired many other researchers to develop new motivation theories [17–22]. Also, the motivation of tourists can be limited by numerous barriers that decrease their willingness to visit a particular destination [8,23–26]. According to Fodness [27], motivation is represented as a driving force that stimulates tourist's behaviour. Iso-Ahola [28] stated that motivation is often used as a predictor of travel behaviour.

Tourists express different behaviour patterns before, during, and after the trip, and tourist behaviour differs from person to person based on internal and external factors [29,30]. Tourist behaviour represents the action individuals take to arrange their time to satisfy travel needs, as well as to consume tourist products or services, under an array of constraints, such as finances, distance, the impact of family and friends, current interests, or time [31]. Tourist behaviours may change over time as a result of constant emerging information and communication technologies (internet, social networks, mobile devices, and online guides), so it is very useful to constantly investigate tourist behaviour to understand their desires and predict possible actions [32]. This will help stakeholders to tailor tourism products according to market trends. Understanding tourist behaviour may help stakeholders to increase tourist satisfaction and create unforgettable experiences. There is a big literature body dealing with the topic of travel behaviour [33–40], however, there is a limited number of articles investigating the travel behaviour of geotourists.

This article intends to develop geotourists' typology models based on travel motivation and travel behaviour for multiattraction destinations. It is also the first research is Serbia related to geotourism that investigates how motives predict travel behaviour before and during the trip. Additionally, the research also focuses on the travelers' attitudes toward the local community. The potential geotourism market of the Zaječar district in Serbia was investigated to get insight into criteria affecting the visitation of geosites. Also, the respondent's attitude toward the local community was investigated to see how geotourists perceive the role of the local community in geotourism development, as many authors [3,12] emphasised their significance in tourism development in a specific area. Therefore, this research contributes to better understanding of what motivates geotourist, their travel behaviour, and attitudes towards local community, and the results could be helpful to policy-makers, destination managers, and all other stakeholders in the Zaječar district, to better prepare geotourism offers and to make market segmentation according to the current trends and expectations of the geotourism market.

## 2. Literature Review

Tourists who visit geosites have different motivations, preferences, and interests and show different travel behaviour and attitude toward the local community. There are numerous articles dealing with the topic of geotourist segmentation and typology. Hose [41] presented two groups of geotourists in Almeria (Spain) with different preferences: dedicated geotourists and casual geotourists. The first one had dominant educational, scientific, and intellectual purposes for the visit, and the second one had dominant recreational purposes and pleasure, as well as limited intellectual stimulation. According to the segmentation of Hose [41], authors Božić and Tomić [42] segmented visitors to canyons and gorges in Serbia and emphasised two different segments based on their main motive, knowledge of geosciences, and preferences: general geotourists and pure geotourists. Pure geotourists are more familiar with the concept of geotourism, and their main motive is related to geology and geomorphology, while general geotourists are less dedicated to geotourism and their main motives are not related to geology and geomorphology. Tessema et al. [11] presented four segments of tourists to geosites in the southern Lake Tana region in Ethiopia based on benefits sought: activity-nature lovers (leisure purposes, younger travellers, travel in a group, low intent to education, low interest in the destination before travel); cultural

lovers (international tourists, low level of interest in attractions related to geology, lower intent to education); nature-cultural lovers (older tourists, the main motive is leisure, like to travel in bigger groups, their trips are arranged by tour-operators); want-it-all segment (lower education level, like to travel in a small group or alone, non-leisure purposes of visit, their trips are arranged by themselves, higher intent to education). Prendivoj [10] presented two groups of geotourists according to their expectations and experiences: latent geotourists and archetypal geotourists. Latent geotourists consist of two subgroups: geotourist lite (travelling independently, in pairs or small groups, interested in novel experience, main motivation factor is visual aesthetic) and mass geotourists (geology is not the main motive, the most common type of visitors of geosites, travelling in bigger groups, low awareness, low intent to education, desire for novelty). Archetypal geotourists consist of two subgroups: social geotourists (scholars or geoscience enthusiasts, education is a priority, high awareness level, like to inform themselves about the site before the visit) and classic geotourists (generally solitary, individualistic travellers, a strong desire for knowledge). Mao et al. [9] presented the preferences of geotourists from Australia, and the authors noted that geotourists want to increase their knowledge of geological sites, obtain intellectual stimulation, satisfy their curiosity, have memorable experiences, and travel independently rather than participate in group tours. Mehmetoglu [43] introduced individualistic and collectivistic tourists to the Western Fjords of Norway. According to his research, group travellers have collectivistic traits, they do not like surprises and they are not involved in trip organisation. On the other hand, individualistic travellers are very negative about group travellers, and their main motivation is novelty/curiosity, escape/freedom, and personal development. Hurtado et al. [5] segmented visitors from Crystal Cave in Yanchep National Park, Western Australia according to their motivation and experiences: purposeful (very high motivation/positive experience), intentional (high motivation/positive experience), serendipitous (medium motivation/positive experience), accidental (low motivation/positive experience), and incidental (low motivation/negative experience). Vasiljević et al. [12] profiled visitors of the National Park Fruška Gora in Serbia. The study reveals five geotourist segments based on their habits: local community-oriented, environmentally aware, nature-based traveller, ecoresponsible, and Plog psychocentric.

The literature related to geotourist motivation is very limited, and few articles address this topic. However, most of the articles address the motivation of geotourists from Australia or Asia, and a few of them from Europe. Kim et al. [44] investigated the motives of geotourists for visiting Hwansun cave in Korea and presented four motivation factors: escape, knowledge, novelty, and socialisation. Hurtado et al. [5] revealed three motivation factors for visiting Crystal Cave in Yanchep National Park, Western Australia: curiosity, education, and great interest in caves. Allan et al. [6] presented the motivation factors of visitors to Crystal Cave, Western Australia: the sense of wonder, relaxation, knowledge, escape, enjoyment, and friendship. All these motivation factors were related to the geotourist sites outside Europe. However, there are few articles dealing with the motivation of visitors to the geosites within Europe. A recent study about motivation-based segmentation of visitors to a UNESCO Global Geopark, conducted by Amaro et al. [45] revealed that visitors to Arouca Geopark (Portugal) could be divided into four groups: the want-it-all geotourists, the true geotourists, sensation seekers, and the accidental geotourists. This was the first study to segment visitors to a geopark in Europe. Two articles present the motivation factors of Serbian geotourists visiting geosites in Serbia, and they are considered very important for comparing the results with this study. Tomić and Marjanović [8] investigated the motivation factors of Serbian geotourist that visited geosites in the region of the middle and lower Danube in Serbia. The results revealed five different factors: visiting attractions, research and prestige, rest and relaxation, and knowledge and friendship. Antić et al. [46] investigated the motivation of visitors to caves in Serbia and revealed four factors (adventure socialisation, active education, sharing experience, and hedonistic well-being). Several past papers dealt with the issue of what various market segments in different countries are consider important while visiting a geotourism destination. This type of research has been

conducted in India [47], Iran [48], Slovenia [49], as well as in Serbia for different market segments and different geosites [42,50–52].

## 3. Materials and Methods

### 3.1. Study Area

Zaječar district is located in the eastern part of the Republic of Serbia (Figure 1). It includes the territories of the municipalities of Sokobanja, Boljevac, and Knjaževac, as well as the territory of the city of Zaječar. It occupies an area of 3624 km$^2$. The geological framework of eastern Serbia represents a complex set of geotectonic entities that reflect the long-term geological evolution of the region. It is dominated by the Carpatho-Balkanides of eastern Serbia, which are separated from the west-located Dinarides by the Vardar Tethys megastructural zone and are covered in the northwest by the formations of the Pannonian Basin. Geological processes in the past period created various peculiarities suitable for the development of geotourism in Eastern Serbia. This area is specific because in such a small area there are numerous different geosites, like waterfalls, canyons, gorges, caves, pits, springs, and mountines, representing the great geodiversity of the Zaječar district. Also, most of them are protected by the Institute of Nature Conservation of Serbia and listed as highly important geoheritage of Serbia. After the emergence of COVID-19 pandemics, this area has been visited by many tourists, and geological attractions have caought their attention. Some of the most famous geosites are Waterfall Ripaljka, Sesalac cave, the canyon of the Moravica River in Sokobanja; Mountain (Mt.) Rtanj, the spring of Crni Timok river, Bogovinska cave in Boljevac; Bigar waterfall, mountain peak Babin zub, Baranica cave, Korenatac gorge in Knjaževac; the valley of Stanjanska river, Lenovac spring, and Barbaroš cave in Zaječar (Figure 2). These geosites are presented in detail in the research of Bratić et al. [53], showing notable values for geotourism development, especially natural values and values regarding protection.

### 3.2. Sampling Method

The survey was conducted using an online questionnaire (Google Forms) and a classic paper-pencil questionnaire. Online questionnaires provided the possibility of wider geographical coverage, while traditional paper-and-pencil questionnaires were filled out by respondents directly at geosites in the Zaječar district. Also, a classic paper-and-pencil questionnaire was delivered to people who do not know how to use modern information technologies. The online questionnaire was distributed through social networks (Facebook, Instagram, Viber, and e-mail addresses), and 148 people filled it out correctly. The target group included professional and amateur mountaineering associations, societies of nature lovers, and students of geography. The classic paper-pencil questionnaire was filled out by 46 people. A total of 194 valid questionnaires were considered for further analysis. All respondents were informed about the reason for the survey, as well as the fact that the questionnaire is anonymous. The only condition to participate in the survey was that respondents reside in the territory of Serbia. The survey was conducted from May to August 2023.

### 3.3. Questionnaire Design

The questionnaire used for this research consists of four parts. In the first part of the questionnaire, there are questions related to the socio-demographic characteristics of the respondents (gender, age, level of education, amount of monthly income, and marital status). Respondents could choose from several answers, and it was necessary to circle one answer in which they recognised themselves the most. In the second part, the questionnaire refers to the motivation for visiting geosites in the Zaječar district. Tourist motivation has long been represented in the scientific literature, and there is a large number of studies dealing with this topic [5–7,9,54,55]. The questions related to motivation were taken from the mentioned works and modified for research purposes to determine the motives of the potential geotourism market for visiting geosites in the Zaječar district. In the third part,

the questionnaire investigated the respondents' travel behaviour. The questions were taken and modified from previous research conducted by Allan [55] and Vasiljević et al. [12]. The fourth part consists of respondents' attitudes towards the local population at the tourist destination. The importance of this topic was emphasised in earlier research [12,56], from which some of the questions were taken and modified. The respondents expressed their level of agreement and disagreement with the statements of the questionnaire by using a five-point Likert scale, where 1 meant "I do not agree at all" and 5 meant "I completely agree".

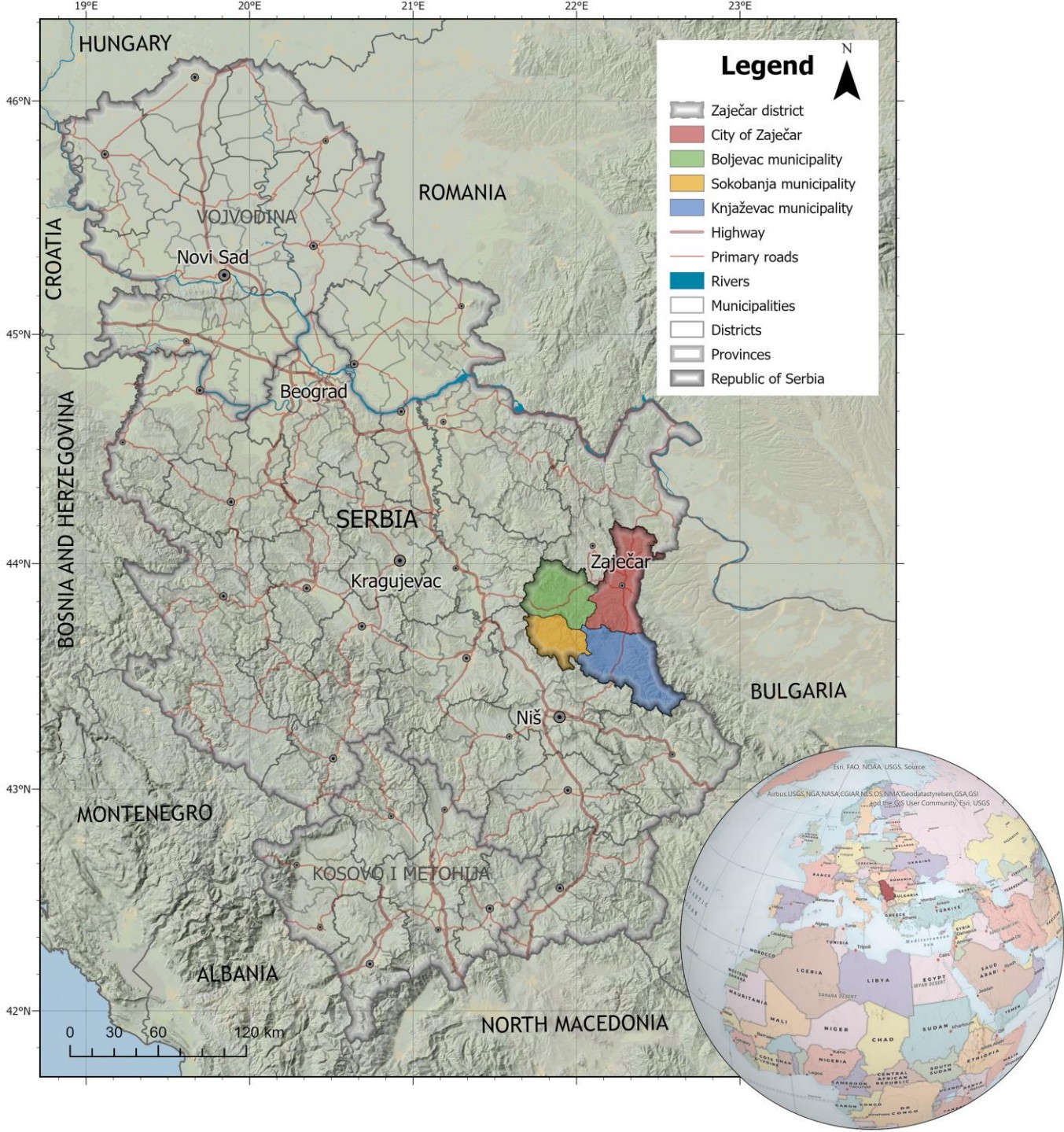

**Figure 1.** Geographical location of Zaječar district within the borders of Serbia.

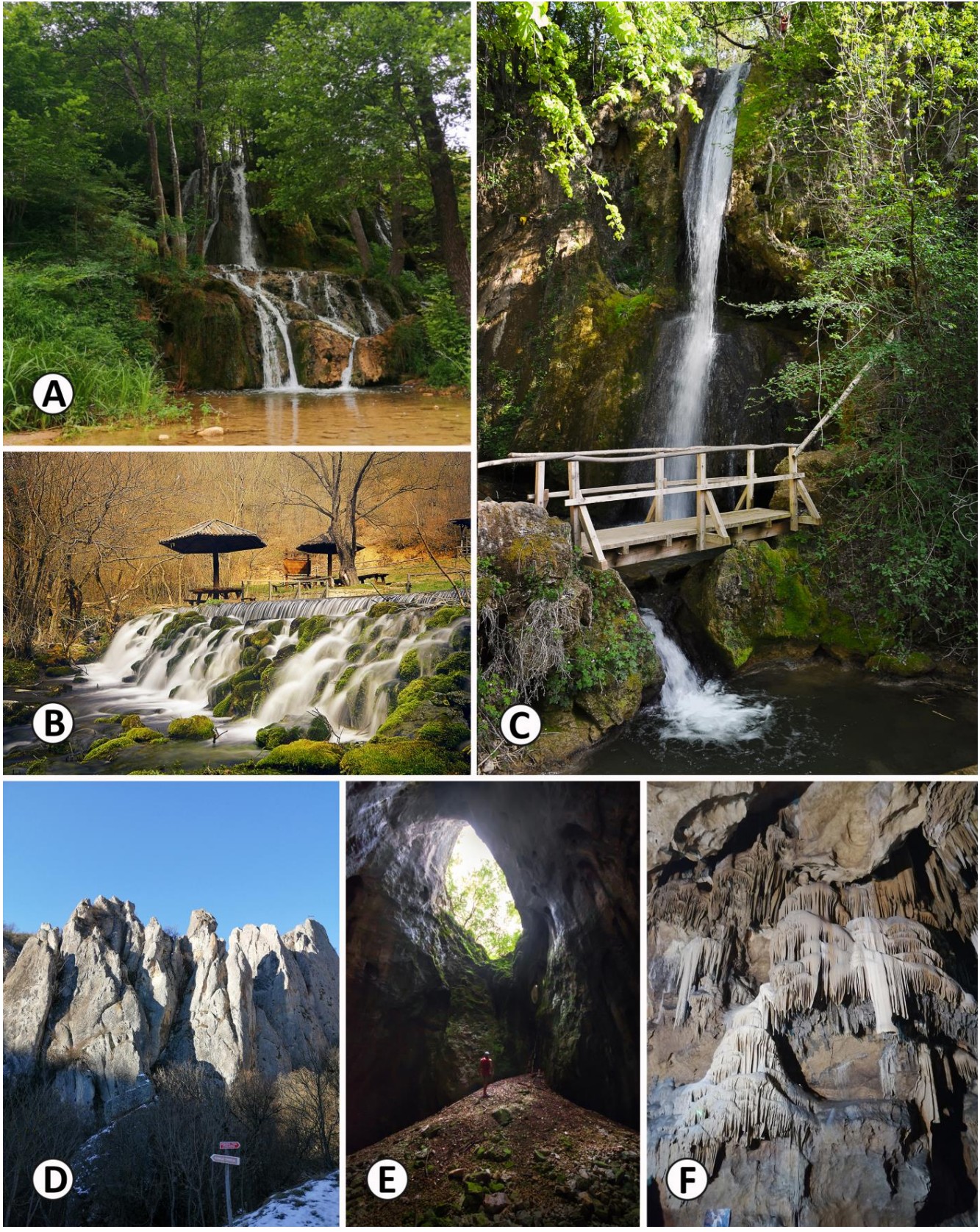

**Figure 2.** Geosites of the Zaječar district: (**A**) Bigar Waterfall; (**B**) the Moravica river spring; (**C**) Ripaljka Waterfall; (**D**) Ždrelo gorge; (**E**) Čitlučka cave; (**F**) Bogovina cave. Photo: Miloš Marjanović (**A**,**C**,**D**,**F**), Bratić et al., [53] (**B**,**E**).

### 3.4. Data Analysis

The collected data underwent thorough analysis using the Statistical Package for Social Sciences version SPSS 2023 (SPSS). To begin, we meticulously examined the socio-demographic characteristics of the sample, utilising a descriptive analysis to gain a comprehensive understanding of the study's participants. Subsequently, we conducted an exploratory factor analysis to categorise and evaluate the potential dimensions of respondents' attitudes towards the motivation for visiting the geosites of the Zaječar district. This analysis also encompassed an assessment of potential dimensions related to travel behaviour and attitudes towards the local community. Our choice of exploratory factor analysis was guided by its proven effectiveness in identifying latent factors within complex datasets, aiding in the comprehensive understanding of nuanced attitudes and behaviours among respondents. Furthermore, we employed one-way ANOVA, the *t*-test for independent samples, and multiple regression analysis to conduct in-depth investigations and ensure robust statistical validation of our findings. Two multiple regression models are set up in the research. The first regression model implies motivation factors as predictor variables, while travel behaviour factors are placed in the position of criterion variables. The second regression model implies that the travel behaviour factors and motivation factors are set up as predictor variables to see how they exert an effect on attitudes towards the local community, which are set in the role of criteria. The selection of these statistical tools was deliberate, as they are widely recognised and accepted methods in social science research for discerning significant differences, correlations, and predictive relationships within diverse datasets.

## 4. Results

### 4.1. Sociodemographic Characteristics of the Respondents

The sociodemographic characteristics of the respondents are presented in Table 1. The majority of the respondents are female (65.98%), while slightly more than a third of the respondents are male (34.02%). More than half of the respondents (54, 63%) are between 31 and 40 years old. Only 17.52% have a high school as their highest level of education, while the rest of the respondents have higher education. Most respondents have completed master's studies (42.79%). Analysing the marital status of the respondents, it can be observed that a larger share of the respondents have a partner (51%), slightly more than half of the respondents. On the other hand, a slightly smaller percentage of respondents are without a partner (49%).

**Table 1.** Sociodemographic characteristics of the respondents (*n* = 194).

| | Number of Respondents (*n* = 194) | % |
|---|---|---|
| Gender: | | |
| Male | 66 | 34.02% |
| Female | 128 | 65.98% |
| Age: | | |
| 16–20 | 5 | 2.58% |
| 21–30 | 53 | 27.32% |
| 31–40 | 106 | 54.63% |
| 41–50 | 15 | 7.74% |
| 51–60 | 8 | 4.12% |
| over 60 | 7 | 3.61% |
| Education level: | | |
| Elementary school | 0 | 0.00% |
| High school | 34 | 17.52% |
| Faculty | 58 | 29.89% |
| Master | 83 | 42.79% |
| PhD | 19 | 9.80% |
| Monthly income: | | |

**Table 1.** *Cont.*

| | Number of Respondents (*n* = 194) | % |
|---|---|---|
| Up to 350€ | 30 | 15.46% |
| 351–550€ | 51 | 26.28% |
| 551–750€ | 46 | 23.72% |
| 751–1000€ | 30 | 15.46% |
| Over 1000€ | 37 | 19.08% |
| Partner (marital) status: | | |
| With partner | 99 | 51% |
| Without partner | 95 | 49% |

### 4.2. Respondents' Motivation

The exploratory factor analysis was applied to see how the motives that influence the decision to visit the geosites of the Zaječar district are distributed. The results of the analysis are shown in Table 2. The factor structure of motives indicates the existence of three factors: The first factor refers to health motives, both psychological and physical in nature. Therefore, this factor (six items) is called *Health and relaxation.* It explains 24.22% of the variance of the motive, and its reliability is $\alpha = 0.77$. The respondents from this group are looking for spiritual calmness, as well as gaining good physical shape in a natural environment. They are motivated by clean and fresh air, and they want to enjoy the positive influence of the sun and geoenergy. Furthermore, spending time in a natural environment encourages them to apply healthy habits every day, and they feel happier during the trip. The second factor gathers items related to the acquisition of knowledge and openness to new destinations and is called *Education and curiosity.* This factor (five items) explains 12.99% of the motive variance, and its reliability is $\alpha = 0.60$. The respondents from this group are constantly seeking to gain new knowledge, especially related to geosciences, and they are not afraid to experience new destinations, attractions, and culture. The last factor refers to the need for company and group affiliation during travel, and this factor is called *Socialisation*. This factor (three items) explains 11.33% of the motive variance, and its reliability is $\alpha = 0.62$. The respondents from this group like to spend their time during the trip with people with similar interests, as well as with their family and friends. They are also open to making new friendships and sharing their experiences. In total, the factors explain 48.54% of the variance, and the factorability coefficients (KMO = 0.65, Bartlett's test $\chi = 649.90$, $p < 0.01$) indicate that the data are moderately factorable.

**Table 2.** Factor analysis of motive structure.

| | Factors | | |
|---|---|---|---|
| | **1** | **2** | **3** |
| Staying in nature in clean and fresh air has an extremely good effect on my health | 0.763 | | |
| This form of tourism makes me happier | 0.735 | | |
| I want to use the positive influence of the sun and geoenergy | 0.596 | 0.328 | |
| Going to nature encourages me to apply healthy habits every day | 0.602 | | 0.460 |
| To relieve my mental state | 0.581 | 0.377 | |
| To refresh my physical condition | 0.790 | | |
| To escape from everyday responsibilities | | 0.543 | |
| To learn new things | | 0.680 | |
| To gain knowledge in the field of geosciences | | 0.673 | |

**Table 2.** *Cont.*

| | Factors | | |
|---|---|---|---|
| | 1 | 2 | 3 |
| To discover new tourist destinations | | 0.540 | |
| To expand my knowledge | | 0.464 | |
| To spend time with family and friends | | | 0.723 |
| To meet people with similar interests | | | 0.734 |
| To make new friends | | | 0.546 |

*4.3. Respondents' Travel Behaviour*

To group the respondents' travel behaviour, exploratory factor analysis was performed. The results identified three factors around which the answers are grouped (Table 3). According to the Scree criterion, the first factor refers to active participation in the process of travel organisation (such as getting details about tourist destinations before arrival, way of transportation, duration of the trip, attractions, excursions, pros and cons), as well as activities in nature (hiking, tracking, riding a bicycle, rafting, kayaking, canyoning, and caving), and the first factor is called *Active behaviour*. Also, this group prefers to spend most of the time in the untouched natural environment of the destination. This factor (five items) explains 24.14% of the variance of the respondents' behaviour, and the reliability of this factor is $\alpha = 0.68$. The second factor refers to the preferences for familiar content, less risk in the organisation of the vacation, as well as a more passive role during the travel organisation, and this factor is called *Passive behaviour*. This group does not have an adventurous spirit and does not create trails by themselves, as they do not want to take a risk, so they are less involved in the organisation and prefer well-known trails. They want to receive information via interpersonal communication rather than via other media. Furthermore, they want to spend their holiday relaxing rather than participating in some kind of activity or learning new things. This factor (three items) explains 17.78% of the variance of the respondents' behaviour, and the reliability of this factor is $\alpha = 0.66$. The last factor refers to *Individual behaviour*. It contains two items related to independence for organising travel itineraries and as little group cohesion as possible when travelling. This group does not want to travel in a big group as they have their tempo, preferences, and dynamics, and they like to adapt the itinerary accordingly. This factor (two items) explains 12.38% of the variance of the respondent's behaviour, and the reliability of this factor is $\alpha = 0.33$, which is much lower than satisfactory because the factor consists of only two items. In total, the factors explain 54.3% of the variance, and the factorability coefficients (KMO = 0.63, Bartlett's test $\chi = 324.26$, $p < 0.01$) indicate that the data are moderately factorable.

*4.4. Attitudes of the Respondents toward Local Community*

By applying exploratory factor analysis, all attitudes are grouped around one factor, and only one factor was singled out, named *Attitudes of respondents towards the local community*. Based on the obtained results (Table 4), it can be seen that the mean values are relatively high, which indicates that among the respondents there is a developed awareness of the importance of the local population at the destination and its participation in the development of (geo)tourism. The claim that the local population should have an advantage in employment in a certain destination (M = 4.50), as well as the claim that tourism in a certain area develops identity and instills pride in the local population (M = 4.41), received the highest mean score. They also believe that tourism should bring employment opportunities and higher income for the local population. The respondents also prefer locally crafted souvenirs, local cousins and restaurants as well as private accommodation owned by the local community. Low values of the standard deviation indicate smaller differences in the answers, and the majority of respondents agree with the given statements.

**Table 3.** Factor sctructure of tourist's behaviuor.

| | Factor | | |
|---|---|---|---|
| | **1** | **2** | **3** |
| Before travelling, I like to get detailed information about the destination I'm visiting (attractions, excursions, how to get there, trip duration, pros and cons of the destination...) | 0.719 | | |
| I like to spend most of my time in nature during my trip | 0.699 | | |
| Natural rarities and beauties are the basic components of the tourist experience on the trips I take part in | 0.699 | 0.325 | |
| I prefer to visit nature in its original form, rather than a modified natural tourist attraction | 0.604 | | 0.331 |
| I like to do sports and recreational activities during my travels (hiking, tracking, kayaking, caving, canyoning, rafting, cycling...) | 0.532 | | |
| During the trip, as well as on the site, I prefer expert guides as sources of information, rather than other sources (internet, information boards, prospectuses, maps...) | | 0.770 | 0.394 |
| I rather prefer to spend my trip relaxing than educational | | 0.728 | |
| I prefer to stick to existing tourist itineraries, and rarely create new itineraries on my own | | 0.618 | |
| I prefer to organise my trip independently according to my own wishes and preferences rather than having someone else do it (a travel agency or a third party) | | | 0.880 |
| I prefer to realize my trip in as small group as possible, or individually, as I want to feel free to travel on my own way and dynamics | | | 0.362 |

**Table 4.** Basic characteristics of individual items and the factor structure of respondents' attitudes towards the local community.

| Variables | Mean | Std. Deviation | Factor 1 |
|---|---|---|---|
| The local community must have a stake in the planning and management of tourism development in a certain area | 4.31 | 0.832 | 0.743 |
| I believe that the income from tourism should be shared by the local community | 4.37 | 0.752 | 0.715 |
| Tourism in a certain area develops an identity and pride in the local population towards its surroundings | 4.41 | 0.744 | 0.714 |
| The advantage of employment in tourism at a certain destination must be given to the local population | 4.50 | 0.853 | 0.703 |
| When choosing accommodation at the destination, I give preference to private accommodation facilities that are owned by the local population | 3.54 | 1.078 | 0.582 |
| I like to try traditional dishes and flavours offered by local restaurants | 4.00 | 1.058 | 0.753 |
| Whenever I can, I buy souvenirs and handicrafts offered by local people | 3.86 | 1.031 | 0.725 |

*4.5. Descriptive Analysis of Variables Related to the Tourist's Behaviuor, Attitudes toward Local Community and Their Motives*

Examining gender differences (male and female) through the *t*-test for independent samples, it is found that gender differences can be identified in the case of *Active behaviour*

(t = 2.73, $p < 0.01$) and *Individual behaviour* (t = 2.87, $p < 0.01$), in both cases in favour of men. Therefore, men, compared to women, are more inclined to take an active role when organising the trip. Also, they want to spend more time in a natural environment and they enjoy activities in nature. Furthermore, they are less oriented towards group cohesion, and rather show individual behaviour. No gender differences were found in the other variables.

In the following paragraph, possible connections (correlations) between age structures, level of education and earnings of respondents with the obtained variables will be presented (Table 5). By looking at Spearman's correlation coefficients, it can be established that age has no significant relationship with the variables included in the research. The level of education is positively related to *individual behaviour*, which means that the more educated people are, the more they will prefer to spend their holidays individually or in smaller groups. Lower educated people, on the other hand, do not perform *individual behaviour*. They want to be a part of a larger group of people during the trip, and they do not want to take part in travel organisation, as they do not want to take a risk. The level of earnings correlates positively with *passive behaviour* and *education and curiosity*, and negatively with *active behaviour*, so people who earn more will be oriented towards enjoyment, resting and relaxing, and they prefer less activity on their vacation. They enjoy doing nothing on their vacation but resting in the natural environment.

**Table 5.** Correlations between age, education level and earnings with research variables.

|  | Age | Level of Education | Amount of Earnings |
|---|---|---|---|
| Active behaviour | 0.01 | 0.04 | −0.13 * |
| Passive behaviour | 0.05 | −0.07 | 0.18 * |
| Individual behaviour | 0.01 | 0.23 ** | −0.10 |
| Health and relaxation | −0.09 | −0.04 | −0.12 |
| Education and curiosity | 0.07 | 0.01 | 0.13 * |
| Socialisation | 0.12 | −0.03 | 0.01 |
| Attitudes towards local. community | 0.05 | 0.06 | 0.08 |

\* $p < 0.05$. \*\* $p < 0.01$.

To examine the potential relationship between partner status and the obtained variables, a *t*-test for independent samples was applied. The results of the *t*-test for independent samples presented statistically significant differences in partner status when talking about certain variables in the research. People who have a partner report that their motives for *health and relaxation* (t = −1.96, $p < 0.05$), as well as *education and curiosity* (t = −2.52, $p < 0.01$) are more pronounced than people who do not have a partner. Likewise, people who have a partner report more pronounced attitudes towards the local community (t = −4.01, $p < 0.01$), as well as that *socialisation* is a more important motive for them (t = −2.36, $p < 0.05$) compared to people who do not have a partner. On the other hand, respondents without a partner expressed dominant *individual behaviour* during the trip (t = 2.74, $p < 0.01$) compared to respondents who had a partner.

The possible relationship between motives and tourist behaviour during travel was examined by a multiple regression analysis (Table 6). By looking at the results of the multiple regression analysis, in which motives were predictors, and tourist behaviour was the criteria, it is pointed to the significant contribution of motives to the manifestation of tourist behaviour during the trip. *Active behaviour* is significantly determined (34%) by motives for travel, especially by the motives of *health and relaxation* in a positive direction. Furthermore, people for whom the motive of *health and relaxation* is the primary motive will express *active behaviour* during the trip. On the other hand, people whose dominant motives are *education and curiosity*, as well as *socialisation*, will express *passive behaviour* during the trip. Moreover, *passive behaviour* is determined by these predictors in a positive

direction (7%). In the end, *individual behaviour* (5%) is determined only by the motive of *education and curiosity*, in a positive direction. That means that people who have dominant motives for education and curiosity will be more guided by personal habits in planning and organising the trip.

**Table 6.** The contribution of travel motives to travel behaviour.

|  | Active Behaviour $F = 33.04$, $R^2 = 0.34$, $p < 0.01$ | | Passive Behaviour $F = 5.03$, $R^2 = 0.07$, $p < 0.01$ | | Individual Behaviour $F = 3.15$, $R^2 = 0.05$, $p < 0.05$ | |
|---|---|---|---|---|---|---|
|  | β | *t* | β | *t* | β | *t* |
| Health and relaxation | 0.66 | 9.08 ** | 0.12 | 1.41 | 0.08 | 0.94 |
| Education and curiosity | −0.01 | −0.08 | 0.36 | 3.82 ** | 0.24 | 2.51 * |
| Socialisation | −0.16 | −1.67 | 0.32 | 2.92 ** | −0.08 | −0.69 |

* $p < 0.05$. ** $p < 0.01$.

By looking at the results of the multiple regression analysis, in which motives and travel behaviour were predictors, and attitudes towards the local community were the criterion, significant parameters of the regression model were obtained (Table 7). These attitudes can be significantly predicted through selected predictors (44%). *Health and relaxation, education and curiosity*, as well as *active behaviour* significantly and positively predict attitudes towards the local community. Therefore, people whose primary motives are focused on education, relaxation, and socializing and those who are health-oriented, as well as those who take an active role in organising their activities during the trip, will have more positive attitudes towards the local community and their involvement into the decision-making process for tourism development.

**Table 7.** The contribution of motives and travel behaviour to attitudes towards the local community.

|  | Attitudes towards the Local Community $F = 24.19$, $R^2 = .44$, $p < 0.01$ | |
|---|---|---|
|  | β | *t* |
| Health and relaxation | 0.43 | 5.19 ** |
| Education and curiosity | 0.39 | 5.05 ** |
| Socialisation | −0.11 | −1.18 |
| Active behaviour | 0.19 | 2.68 ** |
| Passive behaviour | 0.01 | 0.01 |
| Individual behaviour | −0.02 | −0.38 |

** $p < 0.01$.

## 5. Discussion

The exploratory factor analysis was used to group respondents' travel behaviour and motivation for visiting geosites in the Zaječar district. We found three motivation factors (health and relaxation, education and curiosity and socialisation), and three behaviour models (active behaviour, passive behaviour and individual behaviour).

By ranking the motivation factors, we found that *health and relaxation* is the factor that motivates the respondents the most (M = 4.18; SD = 0.70). This might be because of the fact that Sokobanja municipality is one of the most famous spa centres in Serbia, with a lot of healing/relaxing factors (thermo-mineral springs, aero therapy and heliotherapy). Most of the geosites are located near those factors, so this can only fulfil the experience and enjoyment. Also, geosites like waterfalls, canyons and gorges are hydrological by type of

the geosites, and the sound of water falling off the sections and cascades may positively affect inner human senses and relax the visitors. Mountine (Mt.) Stara and Mt. Rtanj dominate in Knjaževac municipality, Boljevac municipality and Sokobanja municipality, so they provide the opportunity to spend time in a natural environment. Rapid lifestyle, which is common in urban areas affects human well-being and forces them to visit destinations with preserved natural environments [3], so this motive could be significant for visiting geosites of the Zaječar district. Nature-based environments may have a positive impact on physical and mental health [57]. The second factor that motivates respondents the most is *socialisation* (M = 3.90; SD = 0.57). This motive has an anthropogenic aspect for visiting geosites in the Zaječar district, and it provides an opportunity to meet new people and cultures, as well as to make new friendships. This could be beneficial for the local community as they can strengthen and promote local identity, offer local products (food, accommodation, rental of hiking equipment, bike rental, etc.) and thus contribute to the local economy and achieve long-term cooperation. The factor *of education and curiosity* motivates the respondents the least (M = 3.84; SD = 0.53). As geotourism is an educative form of tourism, it is obvious that education is an important thing for potential geotourists. Zaječar district has many attractive geosites that can provide educational information for tourists and by explaining certain natural phenomena, the tourist's need to acquire new knowledge about geosciences can be satisfied. Acquainting geotourists with local history and customs is important for localities that have archaeological (Felix Romuliana) and anthropological values (Sokograd), as well as for localities in their vicinity because in this way geotourists are allowed to see the locality through the eyes of the local population, as well as to understand its importance for the local community.

For comparing the obtained results with the results of other studies, two recent articles regarding the motivation of geotourists in Serbia are consulted. Tomić and Marjanović [8] investigated the motivation of visitors to the geosites of the Middle and Lower Danube in Serbia. By applying factor analysis, five factors that motivate visitors were obtained (visiting attractions, research and prestige, physical and mental rest, acquiring new knowledge, and friends). Although the time and space framework, as well as the structure of the questionnaire, are different from the questionnaire that was used in this study, we can conclude that the three motivational factors from the study conducted by Tomić and Marjanović (physical and mental rest, acquiring new knowledge, friends) are quite similar to motivational factors that are emphasised in this study. Antić et al. [46] also segmented the motives for visiting show caves in Serbia and obtained the existence of four factors (adventurous socializsation, active education, sharing experience, and hedonistic well-being). Although the time and spatial framework of this research, as well as the structure of the questions, are different compared to the questionnaire used in this article, the results obtained are quite similar. The three motivational factors obtained from the research of Antić et al. [46] (adventurous socializsation, active education and hedonistic well-being) are similar to the motivational factors obtained in this study. Both mentioned studies offered more motivational factors (Tomić—5 factors; Antić—4 factors), but they are more or less similar to the motivational factors that have been emphasised in this study. Generally, the desire for education, getting to know new and unknown things, making new friends and getting to know different cultures, as well as the desire to maintain a healthy body and spirit, are the motives that dominate the geotourism market in Serbia.

This study also proposed three models of geotourists based on their travel behaviour: active behaviour, passive behaviour and individual behaviour. The model of active behaviour in this study is presented as the willingness of the tourist to take an active role in the decision-making process (destination choice, way of transportation, itinerary etc.), and willingness to take part in the activities in a natural environment (walking, hiking, cycling, rafting, canyoning, etc.) The similarities of such behaviour were noticed in the research conducted by Tessema et al. [11], where geotourists were segmented based on their travel habits, and one of the segments indicated geotourists who like to participate in different activities in a natural environment (activity-nature lovers), such as hiking, caving,

horseback riding, cycling, and mountain climbing, which is similar travel behaviour to the active group of geotourists in this research. Also, Hose [41,58] noted that most geotourists are very active in their travels. Geotourists with active travel behaviour from this study like to get detailed information about the destination they are visiting before arrival. The similarities were found in the research of Božić and Tomić [42] with the pure geotourists, and Mao et al. [9] with archetypal geotourists, as they prefer to consult literature and get detailed information about the sites they visit before the trip, and do not depend on the interpretive content on the site.

Geotourists with passive travel behaviour in this study showed a lack of educational interests, a passive role in trip organisation, and sticking with familiar content. Similar characteristics were presented according to their general geotourists (lack of educational interest, resting and relaxing in nature) in the research of Božić and Tomić [42], and mass tourists (preferring familiar content, passive role in the trip organisation) presented by Mehmetoglu [43] where geotourist do not like surprises and spontaneity during travel, so they usually leave the organisation of their trip to other people, mostly travel agencies. This group presents a majority of the visitors on geosites [5,42].

The third category is geotourists with individual travel behaviour, and their main characteristic is that they like to organise the trip by their own needs and wishes, and they like to travel alone. Similar characteristics of geotourists were found in the research of Prendivoj [10], as classic geotourists are presented as geotourists who like to travel alone or in a small group. Also, in the research of Tessema et al. [11], the want-it-all segment is presented as a segment that self-arranges the trip according to their needs and wishes. Furthermore, creativity during travel and movement "off the track" is characteristic of individual travellers, according to research conducted by Mehmatoglu [43].

The results of the research showed that the majority of respondents have a positive attitude towards the local community of the tourist destination and believe that they are very important for the development of tourism in that area. Allan and Shawandasht [59] present similar views of the interviewees in their study, and state that one of the key factors in the development of geotourism in an area is the local population. The positive effects of the development of rural tourism on the local community can be reflected in three aspects: economic, sociocultural, and in terms of nature conservation [60]. Therefore, it is necessary to include the local population in the vicinity of geosites of the Zaječar district in the geotourism development strategy and assign them a significant role in the planning and implementation in the tourism development plan, because most geosites are located in rural areas.

In the research of Vasiljević et al. [12], a segment of geotourists who have a positive attitude towards the local population at the destination is singled out. This market segment believes that the local community should be enabled: to participate in the development of tourism in its area; that revenues from tourism should go to the local community; and that the local population should be employed in the tourism sector. Also, this segment prefers to be served by the local population; to buy souvenirs and local food products; to eat in local restaurants; and to stay in accommodations owned by residents.

## 6. Conclusions

As geotourism popularity grows every day, it is important to understand what motivates geotourists and what is their travel behaviour, to tailor geotourism products that will meet the expectations of visitors and satisfy their needs. This article aims to determine geotourist typology based on their motivation and travel behaviour, as well as to investigate their attitude towards the local community at the destination. After analysing the results of the questionnaire, it was concluded that participants fit into the geotourist models proposed by other authors. Findings indicate that the respondents are motivated to visit the geosites of the Zaječar district with clean and fresh air, and they want to enjoy the positive influence of sun and geoenergy, as well as gain good physical and mental shape in a natural environment (health and relaxation). Also, they are motivated by educational

possibilities, especially geoscientific, and curiosities (education and curiosity), as well as the possibility to spend time with friends and family or to make new friendships (socialisation). Also, findings indicate three travel behaviour patterns: active behaviour, passive behaviour and individual behaviour. These results could be helpful to destination managers and all other stakeholders to encourage potential tourist markets to consume tourist products such as geotourism and to be more competitive in the market. Also, this may help them create an appealing image that will influence the tourists' choices.

However, this study has some shortcomings that could contribute to future research. Since the segmentation was limited to domestic visitors, it is suggested that the future study include international tourists as well, so that the results may be compared.

**Author Contributions:** Conceptualization, writing—original draft preparation, methodology, M.M.; validation A.A., N.T. and T.T.; formal analysis, M.M.; investigation, M.M., N.T., A.A. and T.T.; writing—review and editing, N.T.; visualization, T.T.; supervision, N.T. All authors have read and agreed to the published version of the manuscript.

**Funding:** This research was partially funded by the Provincial Secretariat for Higher Education and Scientific Research of the Autonomous Province of Vojvodina, Republic of Serbia (grant No. 142-451-3141/2022).

**Institutional Review Board Statement:** Not applicable.

**Informed Consent Statement:** Not applicable.

**Data Availability Statement:** The data that support the findings of this study are available from the corresponding author, M.M., upon reasonable request.

**Conflicts of Interest:** Author Aleksandar Antić is an employee of MDPI; however, he was not work-ing for the journal Sustainability at the time of submission and publication.

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
