# Peer review of "Travel Behaviour Insights among Geotourists in Serbia—Case Study of Zaječar District"

_sustainability, doi:10.3390/su152215969_

Round 1

Reviewer 1 Report

Comments and Suggestions for Authors

This paper is to explore what the geo-tourism motivation and travel behavior. This paper is used quantitative research for understanding how the evidences behind the unknown situation in the proposed case study - in Eastern Serbia.

The section 1 – introduction is ok. It has provided enough information for understanding the research background.

The section 2 – the questionnaire of online and paper to survey has used to collect data, and then the194 questionnaires were valid. And the data were analyzed by SPSS 21, ANOVA, t-test, and multiple regression analysis.

The section 3, 4, and 5 have also provided fruitful research results. And the discussions also reflect to the concerned problem. The conclusion also mentioned about the future work and limitation.

Totally, this paper is a well-organized article, its contents are clearly and interesting. I think that this paper is suitable to be considered for the journal. 

Author Response

The authors thank the reviewer for the comments.

Reviewer 2 Report

Comments and Suggestions for Authors

The paper is easy to read and presents interesting findings. However, it still lacks of literature review that can provide more backbone to the research. My suggestion is to add a separate section to discuss the geotourism research, and more in the Serbian context. Additionally, providing a figure that can show the geotourism site will be helpful also for readers who are unfamiliar with the region.

Comments on the Quality of English Language

N/A

Author Response

reviewer: The paper is easy to read and presents interesting findings. However, it still lacks of literature review that can provide more backbone to the research. 

authors: The section about the literature review has been added. 

reviewer: My suggestion is to add a separate section to discuss the geotourism research, and more in the Serbian context.

authors: The text about geotourism research in the Serbian context is added.

reviewer: Additionally, providing a figure that can show the geotourism site will be helpful also for readers who are unfamiliar with the region.

authors:  A new Figure with the most attractive sites is added.

Reviewer 3 Report

Comments and Suggestions for Authors

After reading the manuscript, I have the following recommendations and comments:

- in my opinion, the authors' idea is of interest to both a specialized and a wider audience; I share their position that the potential of geotourism is important and in the same time underestimated;

- the topic is current and important for the strategic focus for the development of tourist destinations and especially for the specified area in Serbia;

- the study is well constructed and structured.

However, I would like to make the following recommendations:

1. regarding the Abstract

- it would be good to specify the methodology chosen to be used;

- the verb tenses should be specified;

I have comments regarding the definitions used. I do not find a definition that the authors have adopted regarding geotourism. It seems to me very important that it be supplemented, since only in this case it is possible to understand what the object and subject of their study are. For me, they are are not well defined. A research thesis is also missing. In this sense, a revision of the Introduction is necessary.

I also believe that further clarification of the methodology is necessary. A description of the methods used is missing, as well as a justification of why these methods are appropriate.

I have made additional comments in the attached file with the manuscript.

Comments on the Quality of English Language

My overall impression of the manuscript is good. The topic is interesting. The title is correct. I would recommend a slight revision of the Abstract and the Introduction, as well as supplementing the methodology.

I have made additional comments in the attached file with the manuscript

Author Response

Reviewer: A research thesis is also missing. In this sense, a revision of the Introduction is necessary.

Authors: The introduction section is rewritten.

Reviewer: 

regarding the Abstract

- it would be good to specify the methodology chosen to be used;

- the verb tenses should be specified

Authors: the text is rewritten accordingly. 

Reviewer: 

I have comments regarding the definitions used. I do not find a definition that the authors have adopted regarding geotourism. It seems to me very important that it be supplemented, since only in this case it is possible to understand what the object and subject of their study are. For me, they are not well defined.

Authors: The definition of geotourism is added

Reviewer: 

I also believe that further clarification of the methodology is necessary. A description of the methods used is missing, as well as a justification of why these methods are appropriate.

Authors: The text about methodology is rewritten.

Reviewer: the sentence needs correction in in terms of the verb tense used

 “The results presented three segments of geotour-12 ists based on their motivation to visit geosites (health and relaxation, education and curiosity, so-13 cialization), and three segments of geotourists based on their travel behaviour…”

Authors: The text is rewritten

Reviewer: again, correction in the English utterance is required regarding the meaning the authors would like to convey to the reader

 “Also this study presented that respondents have positive 15 attitude towards local community, and highlighted their importance for geotourism development.”

Authors:  The text is rewritten

Reviewer: where is described the concept of this survey? 

Authors: The concept of the survey was described in the section 3 (questionnaire design)

Reviewer: let authors pay attention to such repetitions and avoid them

Authors: corrected

Reviewer 4 Report

Comments and Suggestions for Authors

Taking Zaječar District for example, this paper investigated geotourists typology based on their motivation and travel behaviour. Overall, this paper is interesting and impoartant. Before publication, the paper should be amended as follows.

First, in the third paragraph of Section 1, as authors mentioned, this type of research has been done in some other regions. However, the author discusses this issue of the region, so what is the contribution of this paper?

Second, in the last paragraph of Section 1, authors stated that this article conducted this research from a marketing-oriented perspective. The problem here is twofold. On one hand, is the existing research oriented towards the market or tourists? Or something else? The author needs to give the corresponding explanation. That is, why a market-oriented perspective is needed here. On the other hand, the respondent's attitude toward the local community is discussed here, why does the author deal with it this way? What is the internal logic? The author needs to explain that.

Third, regarding Section 2.1, why is this region selected in this paper, that is, what is the typical significance of selecting this research region? The author needs to be explicit.

Fourth, regarding Section 2.4, as authors mentioned, multiple regression analysis is used here. However, what is the detailed model? What are the variables? The author needs to discuss here

Comments on the Quality of English Language

Good.

Author Response

Reviewer: First, in the third paragraph of Section 1, as authors mentioned, this type of research has been done in some other regions. However, the author discusses this issue of the region, so what is the contribution of this paper?

Authors: The sentence about research contribution is added at the end of the Introduction sector.

Reviewer: Second, in the last paragraph of Section 1, authors stated that this article conducted this research from a marketing-oriented perspective. The problem here is twofold. On one hand, is the existing research oriented towards the market or tourists? Or something else? The author needs to give the corresponding explanation. That is, why a market-oriented perspective is needed here. On the other hand, the respondent's attitude toward the local community is discussed here, why does the author deal with it this way? What is the internal logic? The author needs to explain that.

Authors: 

The section about the market-oriented perspective is rephrased.

The section about the significance of the respondent's attitude toward the local community is additionally explained. 

Reviewer: Third, regarding Section 2.1, why is this region selected in this paper, that is, what is the typical significance of selecting this research region? The author needs to be explicit.

Authors: The text is rewritten accordingly.

Reviewer: Fourth, regarding Section 2.4, as authors mentioned, multiple regression analysis is used here. However, what is the detailed model? What are the variables? The author needs to discuss here

Authors: An additional explanation of multiple regression analysis is provided.

Reviewer 5 Report

Comments and Suggestions for Authors

Dear Authors,

I congratulate you on your work. You can find some smallr corrections in the attached file. I would appreciate it if you could also state the occupations of the participants, which I think it is important for the results of the study. If analyzed, statistical results of the relationship between their occupation and their behaviours should also be given. For example, do geography students have a different attitude/behaviour towards ecotourism than other participants? Even though this is not statistically significant, I think it should be mentioned.

Best Regards

Author Response

Reviewer:  I congratulate you on your work. You can find some smallr corrections in the attached file.

Authors: Corrections are applied.

Reviewer: I would appreciate it if you could also state the occupations of the participants, which I think it is important for the results of the study. If analyzed, statistical results of the relationship between their occupation and their behaviours should also be given. For example, do geography students have a different attitude/behaviour towards ecotourism than other participants? Even though this is not statistically significant, I think it should be mentioned.

Authors: Unfortunately, this parameter was not analysed and it was not a part of the survey.

Reviewer:  Abstract should be redesigned in a much more effective and meaningful way. It should include the subject, location, material(s), method and most striking findings of the research, and the results should be emphasized.

Authors: The abstract is rewritten according to suggestions.

Reviewer: What are they? Please mentioned them...

Authors: References are added.

Reviewer: When citing the other studies, please include page numbers, if it is possible.

Authors: The authors will apply Instructions for authors.

Reviewer:Unnecessary encyclopedic information should be removed from the introduction section. It is not necessary to cite studies containing philosophical discussions if they are not directly related to the research itself, for example if they will not be compared in the findings section or if they are not considered in the methods section. The originality of the research, its aims, and the geographical characteristics of the research area should be given in detail.

Authors: The text is rewritten accordingly.

Reviewer: Another map of the world, or at least Europe, should be drawn, indicating the location of Serbia with a smaller scale. The legend should be designed more clearly. Maybe a white background can be placed behind it.

Authors: Thank you for your comment. In our opinion, the original version of the map was quite readable and better in terms of aesthetics. Perhaps the problem was the smaller resolution in the submitted version. Nevertheless, we have modified the map according to the reviewer's comment.

Round 2

Reviewer 4 Report

Comments and Suggestions for Authors

The authors have made appropriate modifications as suggested, and overall, this paper meets the requirements for publication.

Comments on the Quality of English Language

Good

Author Response

Dear reviewer,

I would like to thank you for the comments which improved the article.

The article was reviewed by the professor of English language and literature, and small corrections were made.

Kind regards,

Authors